# Predicting Hepatitis B Virus Infection Based on Health Examination Data of Community Population

**DOI:** 10.3390/ijerph16234842

**Published:** 2019-12-02

**Authors:** Ying Wang, Zhicheng Du, Wayne R. Lawrence, Yun Huang, Yu Deng, Yuantao Hao

**Affiliations:** 1Department of Medical Statistics and Epidemiology, School of Public Health, Sun Yat-sen University, Guangzhou 510080, China; wangy633@mail2.sysu.edu.cn (Y.W.); duzhch5@mail.sysu.edu.cn (Z.D.); huangy279@mail2.sysu.edu.cn (Y.H.); dengyu_du@126.com (Y.D.); 2Department of Epidemiology and Biostatistics, School of Public Health, University at Albany, State University of New York, Rensselaer, New York, NY 12144, USA; wlawrence@albany.edu

**Keywords:** hepatitis B virus, machine learning, prediction

## Abstract

Despite a decline in the prevalence of hepatitis B in China, the disease burden remains high. Large populations unaware of infection risk often fail to meet the ideal treatment window, resulting in poor prognosis. The purpose of this study was to develop and evaluate models identifying high-risk populations who should be tested for hepatitis B surface antigen. Data came from a large community-based health screening, including 97,173 individuals, with an average age of 54.94. A total of 33 indicators were collected as model predictors, including demographic characteristics, routine blood indicators, and liver function. Borderline-Synthetic minority oversampling technique (SMOTE) was conducted to preprocess the data and then four predictive models, namely, the extreme gradient boosting (XGBoost), random forest (RF), decision tree (DT), and logistic regression (LR) algorithms, were developed. The positive rate of hepatitis B surface antigen (HBsAg) was 8.27%. The area under the receiver operating characteristic curves for XGBoost, RF, DT, and LR models were 0.779, 0.752, 0.619, and 0.742, respectively. The Borderline-SMOTE XGBoost combined model outperformed the other models, which correctly predicted 13,637/19,435 cases (sensitivity 70.8%, specificity 70.1%), and the variable importance plot of XGBoost model indicated that age was of high importance. The prediction model can be used to accurately identify populations at high risk of hepatitis B infection that should adopt timely appropriate medical treatment measures.

## 1. Introduction

Hepatitis B virus (HBV) infection remains a major public health concern worldwide, with an average prevalence of 3.61% [1]. In 2015, over 300 million patients were reported to have viral hepatitis globally—of which, approximately 257 million people were HBV-infected. Additionally, approximately 0.65 million deaths per year were due to HBV infection [2]. In China, the disease burden of HBV is a serious concern. An estimated 90 million people in China—approximately 7% of the national population—are chronically infected with the HBV, and 0.33 million people annually die from HBV-related cancers [3]. According to China Center for Disease Control and Prevention (China CDC), the total cost of treating HBV-related diseases was estimated at 80–120 billion RMB (i.e., Chinese Yuan, CNY) in 2015 [4].

In 2016, the World Health Assembly published the Global Health Sector Strategy, calling for the elimination of the threat of hepatitis in humans by 2030, reducing the number of new viral infections by 95% and the number of hepatitis deaths by 65%. Hepatitis B is a major contributor to the epidemic of viral hepatitis, and the primary target for prevention and control [5]. Hepatitis B is an infectious disease, with severe prognosis and no effective way to eliminate the virus in an infected individual. Moreover, if HBV infection results in the development of a chronic disease, there is a risk that the associated harm will be lifelong. Most HBV carriers are without symptoms in the early stages and are often diagnosed during medical examinations, missing the ideal treatment period [6]. There are also hidden hepatitis B carriers who do not know their infection status. Therefore, the early identification of high-risk groups and timely intervention are effective ways to control HBV infection [7].

Currently, China’s government takes great steps towards ensuring enrollment and employment rights for hepatitis B patients. Laws including the Infectious Diseases Prevention Law require that educational institutions and employers cannot screen for HBV for admission of citizens to school and employment. For this reason, challenges occur as a result of limited available information that can assist in calculating the current prevalence of hepatitis B and controlling HBV transmission [8]. Additionally, general HBV screening is neither cost-effective nor practical [9]. Assessing the risk of HBV infection is important for health care providers to identify patients appropriate for antigen testing. Previous studies aimed to prevent and control HBV infection by identifying risk factors including lifestyle, and corresponding vaccine and infectious history [10,11]. However, risk factors may not be fully identified and the risk of HBV infection was not predicted. Predictive models are widely used in the medical field to quantify population risks of certain disease [12]. If individuals at risk of HBV infection could be identified using a prediction model, it would be possible to perform targeted intervention efficiently. However, there remains a gap in knowledge on an early warning model for HBV infection based on large health screening data.

Currently, machine learning technology is an important branch of artificial intelligence and widely used for analyzing medical data. Machine learning can automatically discover and exploit the interactions and nonlinear relationships between variables and improve the accuracy of disease prediction [13]. A study by Weng, et al., reported that machine learning improves the accuracy of cardiovascular risk prediction and increases the number of patients identified who could benefit from preventive treatment [14]. The purpose of the present study was to develop and evaluate models for identifying people who require screening for hepatitis B surface antigen (HBsAg). We applied machine learning methods to select high-risk groups more efficiently. We believe that the development and application of predictive models will provide important information for law makers to distribute limited medical resources more efficiently and effectively.

## 2. Material and Methods

### 2.1. Data Collection

In the present study, data were obtained from a community-based cross-sectional study enrolling 97,173 residents from Guangzhou city and Zhongshan city in Guangdong, China. Stratified cluster random sampling was used to recruit residents from targeted regions between January 2014 and December 2015. The first level of stratification sampling involved Guangzhou city and Zhongshan city. The Yuexiu district in Guangzhou and Xiaolan district in Zhongshan were chosen, which were the second level of stratification. The third level of stratification was the random selection of communities. The contents of the survey included the collection of demographic information, a physical heath examination, and collecting a blood sample. The blood sample was used to test for blood routine and liver function. The study obtained ethics approval from the Human Ethics Committee at Sun Yat-sen University (L2017030). All research participants signed informed consent. Among participants that agreed to the study and provided informed consent, doctors in community health centers (CHCs) collected venous blood aseptically to screen for HBsAg and biochemical tests, respectively, using enzyme-linked immunosorbent assay and velocity method. The serum was separated from the blood by centrifugation and was transported in small vials in an ice-packed box to maintain their temperature at 0–4 °C to the laboratory at Da An Gene Company. 

A total of 33 indicators with the potential to be associated with HBV were included in the analysis, including demographic information, blood routine indicators, and liver function. HBsAg served as an indicator of HBV exposure and presented as the primary outcome (positive or negative). We randomly selected 80% for training and the remaining 20% for testing. Four models were trained, as described in the classification model sections.

### 2.2. Data Preprocessing

The total HBsAg-positive cases were 8034, accounting for 8.27% of all participants. Data such as this are considered unbalanced (the proportion of the normal population is larger). The synthetic minority oversampling technique (SMOTE), which is an oversampling technique proposed by Chawla et al., is of great popularity to address class imbalance by creating synthetic minority class samples [15]. Borderline-SMOTE combines the original SMOTE and boundary information algorithm, which only oversamples the minority examples near the borderline [16]. The borderline minority examples should first be identified from the original dataset, and then used to generate new minority examples before inserting back to the original one in order to achieve data balance. This study used Borderline-SMOTE to overcome class imbalance problems, reconstruct the training set, and then use machine learning to train the classifier.

### 2.3. Classification Models

We developed models for HBV prediction using four machine learning algorithms: logistic regression (LR), decision tree (DT), random forest (RF), and extreme gradient boosting (XGBoost).

LR is a generalized linear regression model that is commonly applied to binary dependent variables or multiple classification variables, which is chosen as a baseline comparison. It has advantages in the interpretation of model results, and implementation with low computational cost, and can directly derive the weight of each predictor [17]. The disadvantage is that it is sensitive to the multicollinearity of independent variables, making it unsuitable for dealing with data imbalance (i.e., the positive rate of 8.27% in our study), and it may provide an under-fitting prediction.

DT is a tree structure used for classification and regression. DT represents the procedure of classifying instances based on features, which can be considered as the set of if-then rules or the probability distribution defined between feature space and class [18]. The main merits of DT are intuitional results and fast computation. The model is built with training data relying on the principle of minimizing the loss function in learning procedure and applied to classifying testing data. However, it is easy for over-fitting to occur and bias for unbalanced data.

RF is an ensemble algorithm based on a decision tree classifier. The learning procedure combines bagging and random feature selection, which add additional diversity to the decision tree model. RF applies the majority of votes over all decision trees to output the final classification result. This can improve the predictive accuracy without increasing the computational complexity, resulting in the ability to predict outcomes for thousands of variables [19]. RF is also insensitive to the assumption of multivariate linearity, providing robust results for missing or unbalanced data.

XGBoost is a distributed gradient boosting algorithm based on classification and regression trees. XGBoost is popular in the fields of machine learning and data mining, revealing excellent judgment and recognition. The basic principle is to weigh the results of multiple decision trees (weak classifiers) as the final output (strong classifier) [20]. XGBoost achieves good control for model complexity by adding regular items to the objective function, which solves the collinearity problem between variables to a certain extent, and prevents the model from over-fitting. In the XGBoost model, the second-order Taylor series is used for the cost function, and the first and second derivatives are used to make the approximate optimization of the objective function closer to the actual value, thereby improving the predictive accuracy [21].

### 2.4. Tuning of Parameters

The use of XGBoost, RF, and DT for prediction requires tuning several parameters or hyper-parameters. We tuned the parameters or hyper-parameters to maximize the mean area under the receiver operating characteristic (ROC) curve (AUC) value computed from the 5-fold cross validation of the training data. Each time the training data is randomly divided into five subsets of the same size, four subsets are used to train the model and another subset is used for verification. After finding the optimal values of the parameters, prediction models are trained using the entire training data set. The performance is evaluated using the test data. Table 1 presents the tuning parameters and values of the final model for predicate HBV infection.

### 2.5. Evaluation Metric

In our study, we use accuracy, sensitivity, specificity, and area under the receiver operating characteristic (ROC) curve (AUC) as metrics to evaluate the performance of the prediction models [22]. The accuracy, sensitivity, and specificity were calculated as follows:$$Accuracy=\frac{TP+TN}{TP+FP+TN+FN}$$
$$Sensitivity=\frac{TP}{TP+FN}$$
$$Specificity=\frac{TN}{FP+TN}$$
where *TP*, *FP*, *TN*, and *FN* denote true positives, false positives, true negatives, and false negatives, respectively.

Accuracy represents the proportion of correctly predicted samples to all predicted. Sensitivity represents the proportion of correctly predicted positive samples to all actual positive ones. Specificity represents the proportion of correctly predicted negative samples to all actual negative ones. ROC curves are plotted to describe the variance on numbers of correctly classified abnormal cases and those of incorrectly classified normal cases as abnormality. The AUC value is used to comprehensively evaluate the model prediction ability [23].

### 2.6. Statistical Analysis

All statistical analyses were conducted using R software version 3.3.5 (R Core Team, Vienna, Austria). Data that was normally distributed was expressed as the mean and standard deviation, and differences between groups were compared using t test. The categorical variables are expressed in terms of frequency (percentage), and the differences between groups are compared using Fisher’s exact probability method. The R packages involved include XGBoost, glm, rpart, random Forest, smotefamily

## 3. Results

### 3.1. Description

A total of 97,173 participants were included in the analysis. Table 2 presents the demographic characteristics and laboratory testing results. Overall, the positive rate of HBsAg was 8.27%, with a mean age of 54.94. Among the study participants, the ratio of male to female was 0.49:1. In total, 47.02% of participants’ highest level of education was primary and middle school, and 69.79% were married. Only 5.12% of participants received the hepatitis B vaccination among those with confirmed hepatitis B vaccination (11.31%).

In total, 77,738 instances were randomly divided to the training set and 19,435 instances were randomly divided to the testing set, resulting in a random partition of 80%/20%. Differences between the training set and testing set by the demographic characteristics and results from laboratory measurements are shown in Table 3. The *p* values of each variable between training and testing sets were greater than 0.05, indicating that the splitting process was random and balanced.

### 3.2. Predictive Accuracy

The performance of the models predicting HBV infection risk is presented in Table 4, and the ROC curve of each model based on the testing set is shown in Figure 1. The AUCs of the LR, DT, RF, and XGBoost were 0.742 (95% confidence interval (95% CI: 0.729, 0.754), 0.619 (95% CI: 0.603, 0.634), 0.752 (95% CI: 0.740, 0.764), and 0.779 (95% CI: 0.768, 0.791). Two machine learning algorithms achieved improvements in discrimination (1.0% for RF, 3.7% for XGBoost) when compared to the LR model. The predictive performance of the combined models after the Borderline-SMOTE sample resampling was further improved compared to the model used alone. The Borderline-SMOTE XGBoost combined model had the best prediction results, with an AUC value of 0.782.

In terms of other measures, the LR model predicted 12,975 cases correctly from 19,435 total cases, with a sensitivity of 68.7% and specificity of 66.7%. Although the accuracy of the DT was higher than that of XGBoost, it can be seen through the confusion matrix that DT had a bias in the classification of minority class. The Borderline-SMOTE XGBoost combined model improved the identification of minority class, with the sensitivity of 70.8% and the specificity of 70.1%. Full details on the classification analysis can be found in Table 5. Using the variables exhibiting the highest coefficients of permutation importance for HBV infection in XGBoost model, the variable importance plot suggested that age was the most important predictor of HBV infection followed by ALT, PLT, AST, ALB, and PCT (Figure 2).

## 4. Discussion

We developed HBV infection risk assessment models based on health examination data of 97,173 community residents using a machine learning method with the goal of determining the optimal model and improving the detection rate of positive HBsAg. Our findings revealed that the Borderline-SMOTE XGBoost combined model outperformed the other models with desirable performance and may help identify individuals in need of HBsAg testing. The combined model of preprocessing samples with Borderline-SMOTE can solve the problem of data imbalance and improve the overall prediction performance of the model. A large proportion of people unaware of HBV infection missed the ideal treatment time, resulting in treatment difficulties and poor prognosis. There is a lack of assessment of patients at risk of HBV risk in clinical settings. Thus, it is necessary to improve the detection probability of HBV infected patient [24]. Therefore, the XGBoost model can be applied to assess the prevalence of HBV in the general population, promote early diagnosis and timely treatment of high-risk groups, and improve the utilization of medical resources, particularly in low resource countries [25].

The use of machine learning algorithms to predict disease risk has gained attention in the biomedical field [26]. In this study, we took advantage of large-scale datasets to identify individuals at high-risk of HBV infection by applying machine learning methods. Our findings yielded important implications for participants, such as that early identification helps to take effective interventions targeting high-risk groups. Additionally, early treatment in the disease process often means better efficacy. Negative results of the predictive model can eliminate the need for HBsAg testing in most of the general population [27]. Our predictive model can be used to improve the positive detection rate of HBV in areas with limited budget and resources.

Secondly, the predictive performance of the prediction models using machine learning methods was significantly different than that of commonly used traditional classification methods. The more commonly used ensemble model RF and the latest boosting method XGBoost were applied in this study, with controls of traditional machine learning model DT and traditional model LR. The top-performing algorithm, Borderline-SMOTE XGBoost, achieved an AUC of 0.782 (95% CI: 0.771,0.793), and overall accuracy of 70.2%, nearly a four percent higher AUC than that of the traditional LR model. Our findings are consistent with results from a previous study [28]. XGBoost can solve the classification bias problem of traditional models in a few categories, and show strong classification prediction performance on unbalanced data.

The variable importance plot of the XGBoost model showed that age was of high importance to predict HBV infection, which was consistent with a previous study [29]. In China, the majority of HBV infection cases are caused by perinatal vertical transmission and childhood infection. We could infer that older patients with hepatitis B who might have a longer infection time had more serious liver damage and greater susceptibility to adverse outcomes. In order to detect and treat hepatitis B patients early, it is important to carry out long-term follow-up and regular examinations. This study also suggests that variables (alanine aminotransferase (ALT), platelet count (PLT), Aspartate aminotransferase (AST), albumin (ALB), and plateletcrit (PCT)) were important predictors for HBV infection [30]. Our results are consistent with findings from other studies. For instance, the levels of PCT and PLT not only reflect the number of platelets, but also indirectly reveal the functional status of the liver [31,32]. Serum AST and ALT levels are important indicators for examining HBV infection, where content increases are closely related to liver disease [33]. Serum ALB levels can reflect liver reserve, especially synthetic function, which is parallel to the severity of liver disease. The above variables can provide clues and reference for further studies exploring the potential factors of HBV infection prediction.

Our findings also have societal benefits. Adopting risk assessment strategies can provide a greater understanding of HBV prevalence and identify the greatest number of patients for antigen testing [34]. Additionally, the risk of transmission for HBV infection to other individuals can be reduced by early diagnosis with subsequent lifestyle modifications. Moreover, earlier treatment in the course of the disease is related to acceptable cost per quality-adjusted life years estimates [35]. Our results are generalizable with other diseases such as diabetes [27] and cardiovascular risk to a certain extent [14], which can easily build identical predictive models using the same machine learning techniques.

Though our study provides new insight on predicting HBV infection using machine learning algorithm. Several limitations must be mentioned. First, the features we included in our model were based on the obtained datasets. There is a chance that potentially unknown relevant features might have been missing. However, this study included 31 variables and considered as many factors of HBV infection as possible. Additionally, although our model was developed using a limited number of algorithms, it also shows certain representativeness, where XGBoost represents the latest boosting method, RF signifies the traditional integration model, and DT represents traditional machine learning model. In our future research, other machine learning algorithms will be considered to improve the prediction accuracy. Third, we were unable to better analyze the variable of hepatitis B vaccine and provide more detailed information due to missing data on the history of the hepatitis B vaccine. Finally, we note that data was from a community-based study in China and data outside the study area was not used for external verification. However, our data volume is large and still has a certain extrapolation.

## 5. Conclusions

This study applied machine learning algorithms to predict the risk of HBV infection for each participant based on health examination data, and evaluated the predictive effects for the models. Our findings revealed that Borderline-SMOTE sample preprocessing and the XGBoost algorithm together can be used for disease risk prediction with high classification accuracy, which could better assist clinical decision making and treatment. This risk assessment model can be used to diminish the need for antigen screening among low- or non-risk individuals. In addition, interventions in high-risk groups are more cost-effective and reduce associated morbidity and mortality. Through regular follow-up, patients with hepatitis B in the general population can be found earlier. The model can also help to develop a medical resource allocation plan, which has important application value and socioeconomic significance [36].

## Figures and Tables

**Figure 1 ijerph-16-04842-f001:**
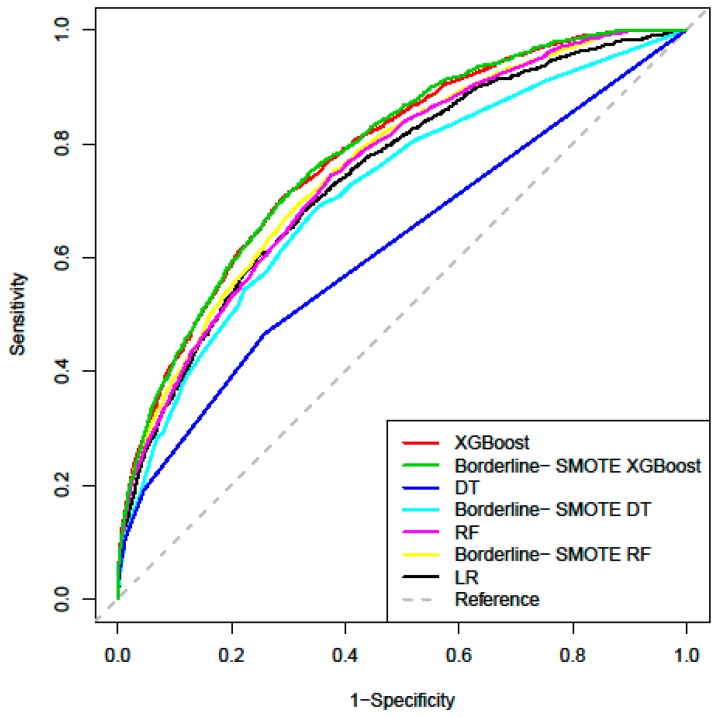
Receiver operating characteristic (ROC) curves of the four models for predicting HBV infection. (XGBoost: extreme gradient boosting; RF: random forest; DT: decision tree; LR: logistic regression; SMOTE: synthetic minority oversampling technique).

**Figure 2 ijerph-16-04842-f002:**
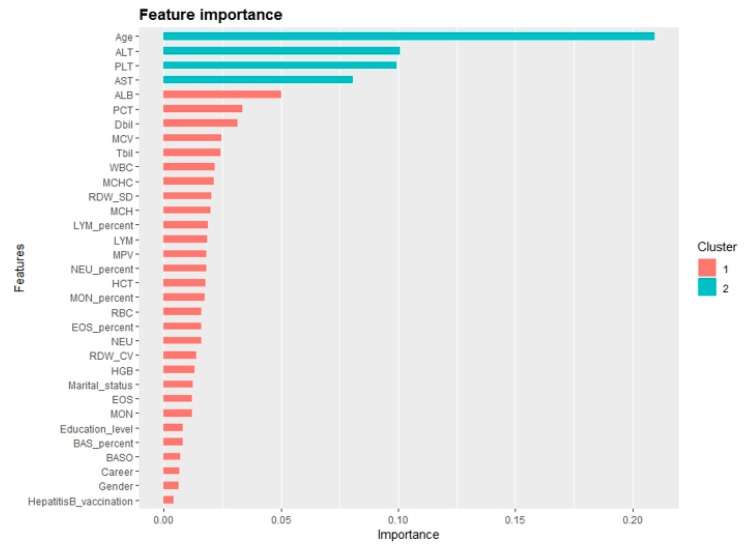
Variable importance plot of the XGBoost model for predicting HBV infection risk. (XGBoost: extreme gradient boosting.)

**Table 1 ijerph-16-04842-t001:** Summary of parameter values in each model for predicting hepatitis B virus (HBV) infection. Decision tree (DT), random forest (RF), and extreme gradient boosting (XGBoost).

Algorithms	Parameter	Value	Meaning
XGBoost	nrounds	120	The number of rounds for boosting.
	max_depth	8	Maximum depth of a tree.
	eta	0.09	Step size shrinkage used in update to prevent overfitting.
	gamma	0.04	Minimum loss reduction required to make a further partition on a leaf node of the tree.
	colsample_bytree	0.8	The subsample ratio of columns when constructing each tree.
	min_child_weight	18	Minimum sum of instance weight (hessian) needed in a child. If the tree partition step results in a leaf node with the sum of instance weight less than the value, then the building process will give up further partitioning.
	subsample	0.89	Subsample ratio of the training instances.
	n_estimators	600	Number of base learners in the integrated model.
	max_delta_step	9	Maximum delta step we allow each leaf output to be. If it is set to a positive value, it can help making the update step more conservative.
DT	minispilt	20	The minimum number of observations that must exist in a node for a split to be attempted.
	minibucket	20	The minimum number of observations in any terminal node.
	maxdepth	10	The maximum depth of any node of the final tree.
	xval	5	Number of cross-validations.
	cp (complexity parameter)	0.001	The minimum improvement in the model needed at each node.
RF	mtry	6	Number of variables available for splitting at each tree node.
	ntree	700	Number of trees to grow.

**Table 2 ijerph-16-04842-t002:** Summary of participants’ characteristics.

Characteristics	*N*/Mean	Proportion (%)/SD
HBsAg		
Positive	8034	8.27
Negative	89,139	91.73
Gender		
Male	32,208	33.15
Female	64,965	66.85
Age	54.94	21.72
Education level		
Illiteracy, and semi-illiteracy	8971	9.23
Primary school	26,024	26.78
Middle school	19,667	20.24
High and vocational school	19,417	19.98
College and above	4632	4.77
Unknown	18,462	19.00
Career		
Leaders of enterprise unit	827	0.85
Technical personnel	2681	2.76
Handle affairs personnel	1844	1.90
Commercial personnel	4768	4.91
Farming, forestry, and fishery producers	7843	8.07
Transportation equipment operators	4430	4.56
Soldier	185	0.19
Unknown	74,595	76.77
Marital status		
Single	16,851	17.34
Married	67,821	69.79
Widowed	4127	4.25
Divorced	821	0.84
Unknown	7553	7.77
Hepatitis B vaccination		
No	6017	6.19
Yes	4976	5.12
Unknown	86,180	88.69
White blood cell count (WBC, 10^9^/L)	6.45	1.75
Percent of monocytes (MON%, %)	4.44	1.87
Monocyte count (MON, 10^9^/L)	0.28	0.14
Red cell volume distribution width-variable coefficient (RDW.CV, %)	14.57	1.38
Red cell volume distribution width-standard deviation (RDW.SD, fL)	55.40	6.91
Red blood cell count (RBC, 10^12^/L)	4.58	0.52
hematocrit (HCT, %)	45.92	4.98
Lymphocyte percentage (LYM%, %)	37.74	9.05
Lymphocyte count (LYM, 10^9^/L)	2.39	0.77
Mean corpuscular volume (MCV, fL)	100.97	10.66
Mean red blood cell hemoglobin content (MCH, pg)	29.55	3.56
Mean corpuscular hemoglobin concentration (MCHC, g/L)	293.22	25.12
Mean platelet volume (MPV, fL)	9.03	0.95
Percent of basophilic granulocyte (BAS%, %)	0.58	0.31
Basophilic granulocyte count (BASO, 10^9^/L)	0.04	0.02
Percentage of eosinophilic granulocyte (EOS%, %)	3.16	2.39
Eosinophil count (EOS, 10^9^/L)	0.20	0.17
Hemoglobin (HGB, g/L)	134.28	14.01
Albumin (ALB, g/L)	45.65	3.27
Alanine aminotransferase (ALT, U/L)	20.68	18.35
Aspartate aminotransferase (AST, U/L)	23.56	13.04
Direct bilirubin (DBil, umol/L)	3.15	1.46
Total bilirubin (TBil, umol/L)	10.39	4.37
Platelet count (PLT, 10^9^/L)	258.25	68.58
Plateletcrit (PCT, %)	0.23	0.06
Percent of neutrophile granulocyte (NEU%, %)	54.08	9.29
Neutrophil count (NEU, 10^9^/L)	3.53	1.32
Total	97,173	

SD, standard deviation. HBsAg, hepatitis B surface antigen.

**Table 3 ijerph-16-04842-t003:** Difference analysis between the training set and the testing set.

Characteristics	Training Set *n* (%)	Testing Set *n* (%)	*p* Value
HBsAg			
Positive	6419 (8.26)	1615 (8.31)	0.812
Negative	71,319 (91.74)	17,820 (91.69)	
Gender			
Male	25,769 (33.14)	6439 (33.13)	0.963
Female	51,969 (66.86)	12,996 (66.87)	
Age(year)	54.90 ± 21.75	55.09 ± 21.64	0.282
Education level			
Illiteracy, and semi-illiteracy	7199 (9.26)	1772 (9.12)	
Primary school	20,855 (26.82)	5169 (26.6)	
Middle school	15,663 (20.15)	4004 (20.6)	0.437
High and vocational school	15,553 (20.01)	3864 (19.88)	
College and above	3666 (4.72)	966 (4.97)	
Unknown	14,802 (19.04)	3660 (18.83)	
Career			
Leaders of enterprise unit	650 (0.84)	177 (0.91)	
Technical personnel	2125 (2.73)	556 (2.86)	
Handle affairs personnel	1463 (1.88)	381 (1.96)	
Commercial personnel	3788 (4.87)	980 (5.04)	
Farming, forestry, and fishery producers	6272 (8.07)	1571 (8.08)	0.633
Transportation equipment operators	3517 (4.53)	913 (4.7)	
Soldier	149 (0.19)	36 (0.19)	
Unknown	59,774 (76.89)	14,821 (76.26)	
Marital status			
Single	13,542 (17.42)	3309 (17.02)	
Married	54,196 (69.72)	13,625 (70.11)	
Widowed	3277 (4.22)	850 (4.37)	0.294
Divorced	674 (0.86)	147 (0.76)	
Unknown	6049 (7.78)	1504 (7.74)	
History of hepatitis B vaccination			
No	4777 (6.14)	1240 (6.38)	
Yes	4016 (5.17)	960 (4.94)	0.229
Unknown	68,945 (88.69)	17,235 (88.68)	
WBC (10^9^/L)	6.45 ± 1.75	6.45 ± 1.73	0.718
MON% (%)	4.44 ± 1.87	4.43 ± 1.88	0.768
MON (10^9^/L)	0.28 ± 0.14	0.28 ± 0.14	0.969
RDW.CV (%)	14.57 ± 1.38	14.56 ± 1.35	0.664
RDW.SD (fL)	55.39 ± 6.91	55.45 ± 6.92	0.239
RBC (10^12^/L)	4.58 ± 0.52	4.58 ± 0.52	1.000
HCT (%)	45.91 ± 4.97	45.97 ± 4.97	0.142
LYM% (%)	37.74 ± 9.05	37.71 ± 9.08	0.616
LYM (10^9^/L)	2.39 ± 0.77	2.39 ± 0.77	0.869
MCV (fL)	100.95 ± 10.67	101.07 ± 10.64	0.157
MCH (pg)	29.54 ± 3.66	29.56 ± 3.13	0.548
MCHC (g/L)	293.26 ± 26.46	293.06 ± 18.85	0.304
MPV (fL)	9.03 ± 0.95	9.03 ± 0.95	0.765
BAS% (%)	0.58 ± 0.31	0.58 ± 0.31	0.146
BASO (10^9^/L)	0.04 ± 0.02	0.04 ± 0.02	0.213
EOS% (%)	3.16 ± 2.39	3.17 ± 2.42	0.736
EOS (10^9^/L)	0.20 ± 0.17	0.20 ± 0.18	0.560
HGB (g/L)	134.26 ± 14.02	134.37 ± 13.98	0.332
ALB (g/L)	45.65 ± 3.28	45.66 ± 3.28	0.731
ALT (U/L)	20.69 ± 19.02	20.62 ± 15.38	0.640
AST (U/L)	23.57 ± 13.50	23.53 ± 11.00	0.696
DBil (umol/L)	3.15 ± 1.47	3.15 ± 1.39	0.632
TBil (umol/L)	10.40 ± 4.39	10.37 ± 4.29	0.448
PLT (10^9^/L)	258.25 ± 68.67	258.27 ± 68.21	0.969
PCT (%)	0.23 ± 0.06	0.23 ± 0.06	0.779
NEU% (%)	54.08 ± 9.28	54.12 ± 9.31	0.610
NEU (10^9^/L)	3.53 ± 1.31	3.54 ± 1.32	0.600
Total	77,738	19,435	

**Table 4 ijerph-16-04842-t004:** Predictive performance of each model for predicting HBV infection risk.

Algorithms	AUC	Standard Error	95% CI	AUC Compared with LR
LR	0.742	0.006	(0.729, 0.754)	-
DT	0.619	0.008	(0.603, 0.634)	−0.123
RF	0.752	0.006	(0.740, 0.764)	+0.010
XGBoost	0.779	0.006	(0.768, 0.791)	+0.037
Borderline-SMOTE DT	0.715	0.007	(0.702, 0.729)	−0.027
Borderline-SMOTE RF	0.759	0.006	(0.747, 0.771)	+0.017
Borderline-SMOTE XGBoost	0.782	0.006	(0.771, 0.793)	+0.040

LR: logistic regression; SMOTE: synthetic minority oversampling technique; AUC: the area under the receiver operating characteristic curve; CI: confidence interval.

**Table 5 ijerph-16-04842-t005:** Summary of evaluation metrics values of each model for predicting HBV infection risk.

Algorithms	TP	FN	TN	FP	Accuracy	Sensitivity	Specificity	Cutoff Point
LR	1109	506	11866	5934	0.668	0.687	0.667	0.010
DT	752	863	13214	4606	0.719	0.466	0.742	0.086
RF	1203	412	11131	6689	0.634	0.745	0.625	0.091
XGBoost	1134	481	12695	5125	0.711	0.702	0.712	0.082
Borderline-SMOTE DT	1094	521	11731	6089	0.660	0.658	0.677	0.135
Borderline-SMOTE RF	1124	491	12121	5699	0.681	0.696	0.680	0.116
Borderline-SMOTE XGBoost	1144	471	12493	5327	0.702	0.708	0.701	0.088

TP: true positives; FN: false negatives; TN: true negatives; FP: false positives.

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
