# Peer review of "Predicting Hepatitis B Virus Infection Based on Health Examination Data of Community Population"

_ijerph, 2019, doi:10.3390/ijerph16234842_

Round 1

Reviewer 1 Report

HBV infection can be easily tested, and the price is not expensive. This predict model is of little significant for clinical practice, while the positive rate of HBsAg was not 8.27% in real world now.

Reviewer 2 Report

The study is interesting, although I think some minor considerations should be corrected:
1.-First, the meaning of the acronym in the text should be clarified, for example, "RMB" at the end of the first paragraph of the intervention.
2.-Secondly, some parts of the material and methods section are difficult to understand. It is difficult for the medical reader to understand the meaning of table 1 and the section entitled "tuning of parameters." This section should be better explained.

Additional comments:

-Abstract: the abstract reflects well the content of the article. It is well written and is complete in its content.

-Introduction: the authors justify in the introduction section the relevance of the study. It should be taken into account that the study is carried out in areas of China where the epidemiology of hepatitis B may be very different from that of other developed countries. Although serology for HBV detection is not difficult to perform or expensive, it may be impossible to reach the entire population at risk of HBV. They justify the relevance of their research to perform targeted intervention efficiently.

-Material and methods: This section is the most difficult to understand. Mainly the section called “Tuning of parameters.” The description of the statistical techniques is somewhat confusing, and the understanding of Table 1 presents great difficulties. I think that this section should be rewritten more clearly, and Table 1 should be explained, mainly if it contributes something important in the communication of the methods used.

- The results are well explained. But despite the effort made by the authors, the predictive parameters selected do not seem to contribute much to new knowledge. Age is the most important variable, but this result seems obvious. It is also true that the sensitivity and specificity achieved with the models are not very high.

-The discussion is well raised.  The model obtained by the authors should be tested with other populations in China. On the other hand, the poor results in specificity and sensitivity, question the validity of the models for their practical application. Perhaps the authors should comment better on these limitations.

Reviewer 3 Report

Ying Wang et al. provide an interesting study on various ML methods in comparison to logistic regression. HBV is a large and continuing health problem, in spite of vaccine availability, so earlier detection etc is a laudable aim. The manuscript is worthy of eventual publication, but there are number of issues to be addressed and corrected, some of particular concern. These are as follows ...

1) "To our knowledge, this is the first study to utilize a large-scale dataset applying machine learning methods to identify individuals at high-risk of HBV infection ...". This is not correct - please see the following publication - Shang et al. Journal of Medical Virology 85:1334–1339 (2013) - DOI 10.1002/jmv.23609. While a smaller total cohort, decision trees and logistic regression were used in tandem to detect and rank top predictors of HBV infection (as modelled against HBsAg);

2) I can see the gist of the modelling, but am I correct to assume that the pathology (and associated) data were modelled, for each ML or LR analysis, against the categories HBsAg POS versus HBsAg NEG?

3) HBsAg "prevalence" was 8.27% - I assume that 8.27% of the total cohort were HBsAg positive?

4) Were the HBsAg POS 8.27% used in the various models directly with the 91.73% HBsAg NEG cases? If yes, this is a significant problem with the analysis, and therefore the veracity of the results. What steps were taken to deal with data imbalance prior to running the ML algorithms? Previous publications have examined this data imbalance problem for HBV + pathology data (see - doi:10.1186/1471-2105-14-206) - why was this, or a similar paper, not cited? I saw mention of data sub-setting in the manuscript, but I'm not clear of how this was applied, and whether it was applied to solve the imbalance problem? If yes, please clarify.

5) How can age, ALT and other leading predictors be used to help increase the efficiency of HBV detection?

6) Some problems with English language expression - "... DT is a popular classification regression method" - do you mean that " ... DT is a popular classification OR regression method? " ... results from laboratory measurements were showed in Table 3" (were shown ...). I have recommended extensive language editing - besides the two examples here, there were many other examples of how clearer language will aid the final paper presentation and legibility.

6) What is "PCT"?

Round 2

Reviewer 3 Report

Thank you for your revised manuscript. As indicated earlier, the Shang et al. paper must be cited (see attached). The manuscript is suitable for publication once this citation update is accomplished.

Author Response

General Comment from authors: We would like to thank the reviewer for their feedback on our manuscript and providing us with further suggestions. We have revised our manuscript following your suggestions.

Comments and Suggestions for Authors

Thank you for your revised manuscript. As indicated earlier, the Shang et al. paper must be cited (see attached). The manuscript is suitable for publication once this citation update is accomplished.

Response: We thank the reviewer for bringing this to our attention. We added the related references. Reference [7] was replaced. (Reference: Line 314-315 of the revised version).